# Incipient Biofouling Detection via Fiber Optical Sensing and Image Analysis in Reverse Osmosis Processes

**DOI:** 10.3390/membranes13060553

**Published:** 2023-05-25

**Authors:** Helge Oesinghaus, Daniel Wanken, Kilian Lupp, Martina Gastl, Martin Elsner, Karl Glas

**Affiliations:** 1Water System Engineering, Chair of Food Chemistry and Molecular Sensory Science, TUM School of Life Science, Technical University of Munich, Maximus von Imhof Forum 2, 85354 Freising, Germany; daniel.wanken@tum.de (D.W.); kilian.lupp@tum.de (K.L.); karl.glas@tum.de (K.G.); 2Research Center Weihenstephan for Brewing and Food Quality, Technical University of Munich, Alte Akademie 3, 85354 Freising, Germany; martina.gastl@tum.de; 3Chair of Analytical Chemistry and Water Chemistry, Technical University of Munich, Lichtenbergstraße 4, 85748 Garching, Germany; m.elsner@tum.de

**Keywords:** reverse osmosis, biofouling, in-situ detection, polymer optical fiber sensors, image analysis

## Abstract

Reverse osmosis (RO) is a widely used membrane technology for producing process water or tap water that is receiving increased attention due to water scarcity caused by climate change. A significant challenge in any membrane filtration is the presence of deposits on the membrane surfaces, which negatively affect filtration performance. Biofouling, the formation of biological deposits, poses a significant challenge in RO processes. Early detection and removal of biofouling are essential for effective sanitation and prevention of biological growth in RO-spiral wound modules. This study introduces two methods for the early detection of biofouling, capable of identifying initial stages of biological growth and biofouling in the spacer-filled feed channel. One method utilizes polymer optical fibre sensors that can be easily integrated into standard spiral wound modules. Additionally, image analysis was used to monitor and analyze biofouling in laboratory experiments, providing a complementary approach. To validate the effectiveness of the developed sensing approaches, accelerated biofouling experiments were conducted using a membrane flat module, and the results were compared with common online and offline detection methods. The reported approaches enable the detection of biofouling before known online parameters become indicative, effectively providing an online detection with sensitivities otherwise only achieved through offline characterization methods.

## 1. Introduction

Water scarcity is increasing, which can be primarily attributed to the effects of a rapidly growing world population, climate change, and increasing industrialization [1]. Consequently, different membrane separation technologies have gained significant research attention in recent years [2]. Among these processes, reverse osmosis (RO) stands out as an effective method for water purification, capable of removing inorganic, organic, and pathogenic pollutants from the feed water [3].

However, a notable drawback of such separation technologies is the formation of deposits on the surface of the membrane. Biological deposits can, for example, pose a significant challenge when raw waters with high organic loads are purified through membrane separation techniques [4]. Microorganisms (MO) tend to adhere to the surfaces within the spiral wound membrane modules (SWM) [5]. Once attached, these MOs rapidly produce extracellular polysaccharides (EPS) and proteins that serve as a protective barrier against cleaning and disinfection agents. Consequently, completely inactivating or cleaning the MOs from the membrane surface becomes nearly impossible [6]. One effective strategy to address the presence of MOs and high organic loads in RO systems is the implementation of pre-treatments that effectively reduce the organic components in the feed water. Therefore, advanced oxidation strategies based on peroxydisulfate are discussed as oxidizing agents that can remove organic pollutants from the water [7]. Alternatively, inactivation of already attached MOs at an early stage can be targeted, specifically before the onset of EPS production. By intervening at this early stage, it becomes possible to rinse the MOs out of the SWM along with organic deposits. As early as 1997, Flemming et al. proposed that cleaning of RO systems should occur before biofouling impacts membrane permeability, before pressure drops in the feed channel, or before membrane biodegradation begins [6].

Hence, a crucial requirement in any anti-fouling strategy is an early in-situ detection of microbial attachment. So far, however, only offline analyses can detect microbial growth in the first attaching and propagation phase. In a recent study, the biofilm formed on the surface of an RO-membrane surface was chemically characterized using ATR–FTIR spectroscopy [4,8]. Another suitable approach to detect a biofilm is to measure the amount of total organic carbon (TOC) [9,10]. A comparison of the TOC values in the feed and retentate allows us to draw conclusions about the accumulated biomass in an RO system. The number of viable MOs is represented by the parameter colony forming units (CFU), which are determined as the cultivable MOs in the water [3]. However, determining the CFUs may be an inaccurate measure due to the presence of clustered cells or non-cultivable MOs in the water [11]. To address this limitation, Vrouwenvelder et al. proposed measuring the concentration of adenosine triphosphate (ATP) as an additional offline parameter, as the ATP concentration is directly proportional to all viable cells present in the characterized waters or biofilms [3,10,11].

However, all offline analytical methods demand significant time-, personnel-, and cost-intensive methods. For instance, the cultivation of MOs according to the German tap-water regulation takes 72 h. Hence, there is a pressing need for developing in-situ analyses of process data and the development of methods that enable biofilm monitoring with sufficient sensitivity to detect the earliest stages of organic and MO accumulation on membrane surfaces [11].

So far, online analyses only indirectly recognize the effects of biofilm formation in membrane processes. Vrouwenvelder et al. proposed a membrane fouling simulator (MFS) be installed prior to the first SWM in an RO plant to additionally measure changes in the feed-channel pressure drop or the permeate flux [10]. This method can detect biofilm formation before the biofouling impacts the SWM of the RO plant [12]. However, an MFS only measures the consequences of biofouling and does not assess the presence of MOs on the surfaces of RO membranes themselves.

In this context, the present study introduces two approaches:A new fiber optical sensor for biofouling detection, which can be easily integrated into both newly constructed and existing SWMs. This sensor provides a reliable method for detecting biofouling in real time within the RO system.The implementation of image analysis techniques for membrane flat modules that are often used in laboratory experiments.

Overall, the approaches present advancements in both in-situ biofouling detection using fiber-optical sensors within SWMs and image analysis techniques applicable to membrane flat modules, thereby enhancing our understanding of biofouling processes in RO systems.

## 2. Materials and Methods

### 2.1. RO-Pilot Plant

The polymer optical fiber (POF) sensors were tested in the same RO pilot plant as previously described by some of us [13]. The only changes made to the current RO setup were the camera position and composition of the water used for the biofouling experiments. The following Figure 1 shows the RO pilot plant used for the present study:

The camera was installed on the feed side of the RO flat module because this is the anticipated location where biofouling should start [10].

Before every experiment, the RO plant was cleaned and disinfected using 0.1 wt% NaOH, 5 wt% citric acid, and 0.25 wt% H_2_O_2_. DOW XLE membranes and a Toray spacer were used for the experiments. The membranes were compacted using a NaCl solution with a conductivity of 200 µS/cm for at least 96 h. Subsequently, the feed water (composition described below) was continuously introduced into the feed vessel.

The feed water used in the experiments consisted of a mixture of Freising tap water (20%) and desalinated water (80%). This composition was chosen to prevent exceeding the calcite solubility product and to inhibit crystal formation. In addition, an autoclaved nutrient solution was added to the feed water to give concentrations (C:N:P) of 500 µg/L, 100 µg/L, and 50 µg/L, respectively. This was achieved by dosing 3.42 mg/L sodium acetate, 1.21 mg/L sodium nitrate, and 0.39 mg/L sodium phosphate into the feed, which had an approximate flow rate of 11.5 L/h. These C:N:P concentrations were half of those used in RO systems previously reported in the literature [3,14]. Throughout the experiment, the retentate and permeate streams were discarded.

### 2.2. POF-Sensor

The POF-sensor principle is presented in Figure 2. Polymer optical fibers consist of a core made of polymethylmethacrylate (PMMA) and a fluoropolymer cladding, which maintains the total internal reflection in the fiber core. In this study, the cladding of the fiber was removed on a length of ca. 5 cm in two different methods: chemical removal with ethyl acetate or mechanical removal using a 350-grid abrasive paper. The removal of the fiber’s cladding enables the interaction of light with deposits on the fiber surface. As recently demonstrated by Hager et al., such a POF sensor is capable of detecting crystals on the fiber surface [13,15]. It is worth noting that the removal of the cladding influences the surface roughness of the fiber. This can be seen in Figure 2, which shows a comparison of scanning electron microscopy (SEM) images of the surface observed after removing the cladding via different methods.

The fibers were installed below the feed spacer or as a spacer substitute. The setup with the incorporated fibers, the feed spacer, and the RO membrane is shown in Figure 3.

### 2.3. Offline Analytic

The following paragraphs describe the offline analysis of the water and the biofilm formation during and after the experiments:

TOC

The analysis of the TOC content as non-purgeable organic carbon (NPOC) was carried out by the Research Center Weihenstephan for Brewing and Food Quality according to DIN EN 1484:1997-08.

Using the TOC concentrations in the feed and the retentate, *TOC_Feed_* and *TOC_Ret_*, respectively, in combination with the flow rates in the feed and the retentate, ${\dot{V}}_{Feed\;}$and ${\dot{V}}_{Ret}$, respectively, the carbon rate ($\dot{C})$ [mg/h] was balanced.
(1)$$\dot{C}=TOC_{Ret}\cdot {\dot{V}}_{Ret}-TOC_{Feed}\cdot {\dot{V}}_{Feed}\;$$

ATP

ATP samples were immediately frozen after sample collection by submersion in liquid nitrogen to avoid the degradation of extracellular ATP. Subsequently, frozen samples were immediately thawed and measured. Measurements of ATP content were conducted using the EnSURE Touch™ (Sundern, Germany) luminometer from Hygiena (Camarillo, TX, USA) using UltraSnap™, Aquasnap™ Total, and Aquasnap™ Free testers in accordance with the manufacturer’s instructions. ATP standards were prepared using adenosine 5′-triphosphate disodium salt from Sigma-Aldrich Chemistry (St. Louis, MO, USA).

CFU

The determination of the colony-forming units of cultivable microorganisms at 22 and 36 °C was carried out by the Research Center Weihenstephan for Brewing and Food Quality in accordance with the norm DIN EN ISO 6222:1999-07 corresponding to § 15 (1c) of the German tap water regulation (TrinkwV).

16S rRNA

Sampling was performed after the end of the experiment by sterile swabs (Texwipe^TM^, Kernersville, NC, USA) directly from the membrane. Samples were processed by ZIEL—Institute for Food & Health, according to Reitmeier et al. [16] with primers 341F/785R. In deviation from the experimental instructions, DNA isolation was carried out after bead beating using the Maxwell^®^ RSC Fecal Microbiome DNA Kit in the MaxWell device from Promega (Walldorf, Germany).

### 2.4. Online Analytic

Feed channel pressure drop (FCP)

The FCP value was measured using Endress &Hauser differential pressure sensors (Type PMD75).

Permeability

The permeability was calculated using the flow rate of the permeate (${\dot{V}}_{p}$) with respect to the net transmembrane pressure $TMP_{net}$ and the membrane area A.
(2)Pw=V˙pA⋅TMPnet 

### 2.5. Image Analysis

A Basler ac-A3088-57uc camera with a CMOS sensor and with 6.4 MP resolution was used to take photos of the membrane surface at 1 h time intervals. The photos were taken in reflected light mode at the feed side of the membrane flat module. The light was turned on for 2 min to avoid algae growth on the membrane surface. The photos were then stored as *.tiff* data with image sizes of 2064 × 3088 × 3 pixels. The terms ‘matrix’, ‘photo’, and ‘image’ are interchangeably used throughout the following text, as they all refer to the same visual representation. The matrices were analyzed using Matlab^®^ R2021a. The Matlab^®^ code is organized into two steps with (A) the image registration and (B) the calculation of image similarities.
(A)Image registration:The motion of the module (due to motor movement etc.) resulted in an imperfect alignment, so the captured images corresponded to different coordinates on the membrane surface. Therefore, it was necessary to perform image registration to align the images. This process followed the four steps of image registration as outlined by Zitová and Flusser [17]: feature detection, feature matching, transform model estimation, image resampling, and transformation.A distinctive section of the photo containing specific features of a cropped reference greyscale image was manually selected.The images were aligned. The Matlab^®^ function *normxcorr2* was used to create a normalized cross-correlation matrix between the selected feature image section and the sensed images of the time series, which were transformed into greyscale images. This function moves the smaller matrix containing the features across the bigger matrix to find the location via the maximum in matching [18]. Next, parameters for further transformation, namely aligning the sensed images around the selected features, had to be extracted using the Matlab^®^ functions *find* and *max*.The gained parameters were then used to align the color images. To be able to transform the images around the same coordinates, they had to be cut in size; thus, gaining room for movement. Hence, the registered images were somewhat smaller than the original ones and consisted of 1937 × 2913 × 3 pixels. As a result of the registration process, all images of the time series had the same size and were centered around the same distinctive features.

The process of image registration is sketched in Figure 4.

(B)Image similarities:

A Pearson correlation coefficient is a very suitable tool to measure pixel-by-pixel image similarities [19,20]. Therefore, analysis of image similarities was carried out via a 2-dimensional Pearson correlation coefficient. Using our image analysis, we determined the different red, green, and blue layers of one RGB image and compared them with a reference matrix, which is the RGB image at the initial time. This photo was taken before the compaction phase ended and the feed water was changed to the tap water desalinated mixture with dosed nutrients.

A correlation coefficient for each sensed image was calculated in the time series and for each RGB layer. The single layers were separated via the Matlab^®^ function *imsplit*, resulting in three 2D grayscale matrices, each representing one RGB layer. Next, a reference picture, matrix ***A*** of size 1937 × 2913, was compared with each sensed image layer, matrix ***B*** of the same size, using the following equation:(3)$$r=\frac{\sum_{m}\sum_{n}\left(A_{mn}-\bar{A}\right)\left(B_{mn}-\bar{B}\right)}{\sqrt{\left(\sum_{m}\sum_{n}{\left(A_{mn}-\bar{A}\right)}^{2})(\sum_{m}\sum_{n}{\left(B_{mn}-B\right)}^{2}\right)}}\;$$
with *A_mn_* as the reference matrix of size *mxn* and *B_mn_* as the sensed matrix of the same size. $\bar{A}$ and $\bar{B}$ are the mean pixel intensities of *A_mn_* and *B_mn_* with *m* = 1937 and *n* = 2913.

The calculated correlation coefficient serves as a measure of the deviation between an image captured at a specific time and the initial reference image, quantifying the level of similarity between the two images.

## 3. Results and Discussion

The results are divided into three experimental parts to demonstrate the function of the POF and the image analysis for early biofouling detection on RO membranes.

### 3.1. Conditioning: POF-Transmissions in Water

The initial set of experiments aimed to demonstrate the long-term effects of POF transmission. The POFs were integrated into an RO system under normal filtration conditions, without forming deposits on the membrane and spacer surfaces. The RO system operated using a 2 g/L NaCl feed solution over the course of several days, with the retentate and permeate flowing back into the feed vessel. Figure 5 illustrates the time series of transmission obtained from these experiments, which were conducted in triplicate.

The recorded time series clearly show an asymptotically decreasing light transmission. The magnitude of the total decrease varied among the three experimental runs. In all experiments, the decrease in light transmission ends after ca. 72 h and a limiting value appears to be reached after 96 h.

Additionally, all three recorded time series exhibit daily variations. These variations are attributed to the daily cycle of temperature fluctuations, which result in changes in the mechanical stresses experienced by the membrane flat module, due to a differential thermal expansion of the metal screws used to compress the module.

These findings are in line with previous reports that have documented a similar influence of water adsorption and mechanical stress on the light transmission of POFs [2,21,22].

### 3.2. First Validation: POF-Sensors in a Yeast Suspension

The second series of experiments aimed to investigate the impact of viable cells and organic suspensions on the light transmission of the POFs.

In contrast to the above-described experiment, the POF was not installed in the RO-flat module but in a water bath, to which a yeast suspension was added after a conditioning period of 71 h. Two fibers were installed in the water bath: a reference fiber with intact fluoropolymer cladding and a sensor fiber with mechanically removed cladding. Figure 6 shows representative results of one of the three experimental runs with this setup.

Figure 6 shows a plot of the recorded transmission over time. The plot shows a nearly constant time series for the transmission of light through the reference fiber over the course of the whole experiment. The plot for the sensor fiber, by contrast, showed a sharp drop in the intensity of transmitted light upon the addition of yeast to the water bath. Hence, the light transmission was significantly influenced by the presence of viable cells. The other two experiments gave similar results.

As previously observed, both time series showed small intensity fluctuations over the course of 24 h due to the daily temperature cycle. The conditioning period ended earlier than in the setup where the POFs were installed in the membrane flat module (see Section 3.1). The yeast was added to a water bath and therefore the POFs were not installed in the membrane flat module, where the screws of the module compress the POFs. The absence of screw pressure (as illustrated in Figure 1) may have facilitated the adaption of the POFs to the experimental conditions.

### 3.3. Second Validation: Biofilm Detection in the RO Pilot Plant

In this following section, the capability of the selected POF sensor and our image analysis approach toward detecting biofouling in RO systems at an early stage is evaluated and compared with current online detection methods.

Prior to the experiments, the POF sensors were conditioned for at least 96 h using a NaCl solution to avoid any effects on the intensity of transmitted light due to water adsorption or mechanically induced transmission loss. The experiment was executed three times. Because the results of all experiments were very similar (see Figure 10), we will focus on discussing the findings from one experiment.

Figure 7 shows the time series of the measured online parameters. The vertical dotted line marks the end of the conditioning and compaction phase; this also marks the moment the dosing of nutrients was started and the switch in feed composition to a mixture of tap water and desalinated water. Figure 7 displays the recorded time series from 96 h onwards, at which point the POF transmission reached a steady-state level (see Section 3.1).

The analysis of the recorded time series shows the effect of biofouling at a very early stage. After the feed water change, the monitored online parameters exhibited changes in the following temporal order: first, the POF’s light transmission was affected; then, the correlation coefficient from the image analysis; subsequently, the permeability; and finally, the FCP value was affected. This temporal order is described in the following paragraphs, assuming that biological growth affected the parameters.

The light intensity transmitted by the POFs on the feed and the retentate side immediately decreased after the end of compaction and with the onset of dosing nutrients to the changed feed solution. The POF’s light transmission on the feed side of the membrane module stopped decreasing and reached a steady-state level from 240 h onwards while the retentate fiber’s light transmission continued to decrease.

The correlation coefficients determined by the image analysis followed a similar trend as the transmission of the POF sensors until 240 h. Subsequently, and nearly simultaneously with the changes in permeability and FCP, the correlation coefficient of the blue layer in the RGB image experienced a significant decrease. This observation could potentially indicate the onset of a phase of exponential growth of a biofilm on the membrane surface. By contrast, the correlation coefficient of the green layer only slightly decreased over the course of the whole experiment, whereas the correlation coefficient of the red layer remained at an almost constant level after the slight decrease following the nutrient dosage. The variations in the correlation coefficients determined for the R-, G-, and B-layers demonstrate that a separate evaluation of the color layers of the images is crucial for detecting biofouling. Furthermore, the differences in the correlation coefficients show the color and brightness shift from white to brown, which were indicative of the growth of a biofilm.

Figure 7 shows that permeability first increased and later (220 h) decreased. The increased permeability can be attributed to pH-dependent membrane surface charges, which changed as a result of the increase in pH from 6.9 to 8 after the compaction, in agreement with previous reports [23,24].

The FCP value was observed to increase after 221 h, with the increase becoming more pronounced over time.

The only parameter that stayed almost at a constant level over the course of the experiment was the salt passage. This indicates the absence of scaling on the RO membrane. The absence of any crystals on the membrane was independently confirmed by SEM imaging (not shown here).

All the observed parameter changes indicate an increasing level of biofouling on the membrane and spacer surfaces. As known from the literature, the MOs adhere first to the feed spacers and therefore to the POFs, which correlates with the initial changes observed in the POF transmission and the image analysis [25]. Moreover, the biofilm started growing on the feed side of the membrane module and later on the retentate side. This phenomenon can be observed from the signal of the two POF sensors. The transmitted light of the POF on the feed side decreases first. The stabilization of the signal at a steady-state level suggests that the biofilm may have completely covered the fiber surface. On the retentate side, the light transmission through the fiber sensor exhibited a more prolonged decrease compared with that of the feed fiber, indicating a delayed biofilm growth on the retentate side.

After ca. 236 h, the biofilm had developed on the membrane surface, which reduced the flux and consequently the permeability. Only a few hours later, the correlation coefficient of the blue layer in the image analysis showed a second drop.

Because the growth of the biofilm was more pronounced in the feed channel, the pressure progressively decreased. In our experiment, the change in pressure was the last parameter to indicate biofouling. The weak increase observed at the beginning of the experiment was not statistically significant compared with its strongly fluctuating signal.

To validate and compare the POF with offline analytics, which are commonly used to indicate and measure microbial growth in RO plants [3,9,10,11], a comparison with the measured offline parameters was carried out, as shown in Figure 8.

The offline parameters measured at different times indicate the different phases of biofilm formation in the membrane flat module.

First, the negative carbon rate and relatively low CFUs indicate the adhesion and growing phase of the MOs. This initial biofouling phase ended after ca. 160 h, when the CFU in the retentate started to increase compared with the feed values. Furthermore, the carbon rate turned to a positive level. Later, after 207 h, the ATP concentration in the retentate increased compared with the ATP concentration in the feed. Notably, however, the ATP value was very low due to the low cell concentration, and it, therefore, remained difficult to precisely measure the ATP concentration.

All the changes observed in these parameters mark the beginning of the second phase of biofouling, the detaching phase, when parts of the biofilm detach from surfaces inside the membrane flat module. At this stage, the MOs, together with the EPS, flow out of the membrane flat module and cause a positive carbon rate and higher contents of viable MOs in the retentate, consequently increasing the ATP concentration. The determined values indicate that this stage reached a steady-state mode when the biofilm inside the RO module reached a constant thickness. This equilibrium was indicated by the ATP concentrations and CFU in the retentate that ceased to increase, corroborating that the mass of growing biofilm and detaching biofilm are balanced.

The comparison of offline parameters with the new online parameters determined via the integrated POF sensors and through image analysis validates the new sensors’ functionality. Simultaneously to the negative carbon rate, the POF signal decreased in parallel with the correlation coefficient from the image analysis. When the second phase began, the image analysis showed a sharp drop in the correlation coefficient of the blue layer, and the POF sensor on the feed side turned to a steady-state level (244 h). The steady-state mode of the retentate fiber from 244 h onward may be caused by a completely overgrown surface of the POF. This observation is in agreement with previous findings [10].

In the final step, the MOs in the biofilm are compared with the MO found in other RO plants known from the literature.

Biofilms are composed of an assemblage of MO at a boundary layer consisting of either a mixed population or individuals of a single species [26]. Typical species found in biofilms on RO membranes belong to ubiquitous aquatic microorganisms. These include organisms of the classes Alphaproteobacteria (for example, order/family/genus Rhizobiales/Bradyrhizobiaceae/Rhodopseudomonas or Sphingomonadales/Sphingomona-daceae/Sphingomonas), Betaproteobacteria (for example, Burkholderiales/Comamonadaceae/Acidovorax), and Gammaproteobacteria (Pseudomonadales/Pseudomonadaceae/Pseudomonas or Xanthomonadales/Xanthomonadaceae/Pseudoxanthomonas) [3,27,28,29]. Identification of MOs can be achieved through various methods, including through culturing techniques (e.g., plate cultures) in combination with Matrix-Assisted Laser Desorption/Ionization Time-Of-Flight (MALDI-TOF) mass spectrometry analysis or by 16S rRNA sequencing, and by other methods. However, MALDI-TOF spectrometry has the disadvantage that cultivability of the organisms is required, which is not always given. On the other hand, 16S rRNA sequencing allows for a fairly accurate estimation of the cultures contained in the biofilm based on highly conserved RNA segments, but no statement can be made about the viability of the detected organism within the biofilm [30]. Here, we chose 16S rRNA sequencing to analyze the species contained in the biofilm.

Table 1 provides an overview of the microbiological composition of the biofilm. Most of the identified microorganisms (85%) belong to Alpha- and Gammaproteobacteria. Most dominant were the bacteria of the class Gammaproteobacteria, which accounted for 58% of the detected bacteria. Typical biofilm formers such as Acidovorax, Pseudomonas, and Pseudoxanthomonas were detected in the forced biofilm formation experiments. The presence of bacteria of the class Bacteroida was also characteristic of biofilms in membrane processes [31].

In all experiments, the bacteria present in the drinking water pipe system led to a bacterial community comparable to that in industrial full-scale RO plants. Hence, our new methods should be suitable for typical biofouling detection in other RO plants.

### 3.4. Practical Test: POF-Sensors as Indicators of Cleaning-in-Place

In the following, insight into possible applications of the POF as a biofouling sensor are provided. The early detection capability of the POF sensors enables the control of a cleaning-in-place (CIP) procedure for membrane modules. This cleaning process is crucial to be carried out before the biofilm significantly impacts the permeability or the feed-channel pressure drop. By implementing effective CIP procedures based on early detection using POF sensors, sustainable desalination through RO membranes can be achieved, even for water sources with high organic loads. Figure 9 shows results from an experiment conducted under accelerated standard conditions, as described in Section 3.3, illustrating the onset of biofouling in this scenario.

Figure 9 shows four time series. On top, the feed channel pressure drop is shown, which has been conventionally used to detect biofouling in spacer-filled feed channels in RO modules. The FCP value remains relatively constant as it is only influenced by the CIP intervals due to Moody’s law.

The lower diagram provides the time series of the new POF sensors. The sensor POF shows the typical descending behavior when biofouling occurs.

Furthermore, a higher crossflow velocity (displayed by$\;{\dot{V}}_{Ret}$) at 128 and 131 h did not influence the POF signals because no nutrients were dosed to the water and the feed water was not yet polluted with microbial load. Because the nutrients were dosed after the compaction and conditioning phase, the POF’s transmission signal began to immediately decrease compared with the reference fiber. At 151 and 178 h, two CIP intervals were executed. In Figure 9, it can be observed that the transmission signal of the POF sensor recovers after each CIP interval. However, the magnitude of the recovery is lower during the second CIP interval. This could be attributed to the presence of increased quantities of the EPS matrix within the biofilm, which provides stability and protection against mechanical cleaning. Prior to the second CIP interval, the signal of the POF sensor reached a steady-state level, potentially indicating a complete coverage of the fiber surface by the biofilm. This observation supports the hypothesis of a stronger EPS matrix within the biofilm during advanced stages of biofouling growth.

## 4. Conclusions

This publication highlights the advantages of combining two novel detection methods for early-stage biofouling monitoring in RO modules. The methods are introduced, their effectiveness is demonstrated, and their benefits are evaluated in comparison with conventional detection methods. By utilizing these new detection methods, it becomes possible to detect and address biofouling at an early stage, enabling more effective management and prevention strategies in RO systems.*Image analysis* can quantify the color changes caused by microbial growth at a very early stage. A preparatory step is needed to adjust the photo’s positions to the reference images recorded during the conditioning phase. A 2-dimensional Pearson correlation coefficient of the R-, G-, and B-layers was calculated for each photograph of the whole experimental series and compared with the reference image. This results in a time series of image analysis parameters that can be recorded while biofouling is affecting the RO process.*Polymer optical fibers* are a new method to detect biofouling throughout the entire growth period. The detection process requires the use of conditioned fibers and enables the qualitative detection of biological growth until the fiber surface is completely covered with biomass. The time series of the transmitted light through the fibers strongly differs from the changes observed in fibers used to monitor scaling (inorganic deposit) on the RO membrane.

Figure 10 provides a visual representation of the early detection and compares the sensor signals that indicate biofilm presence by means of different detection methods. The changes in the time series of the sensor signals occur at similar points in time. Image analysis and the POF’s light transmission indicate the biofouling formation first.

It should be noted, however, that the exact mechanism behind the POF sensors and their detection of biological growth remains uncertain. It is unclear whether the light transmission is influenced by changes in evanescence or if the light is scattered out of the fiber due to inhomogeneities in the biofilm and resulting refractive index changes.

Further research is needed, especially with fiber optic sensors based on fluorescence measurements of organic pollutants on the fibers and feed spacers. This may lead to a species-sensitive measurement of forming biofilms due to the different metabolism products of various bacterial species.

## Figures and Tables

**Figure 1 membranes-13-00553-f001:**
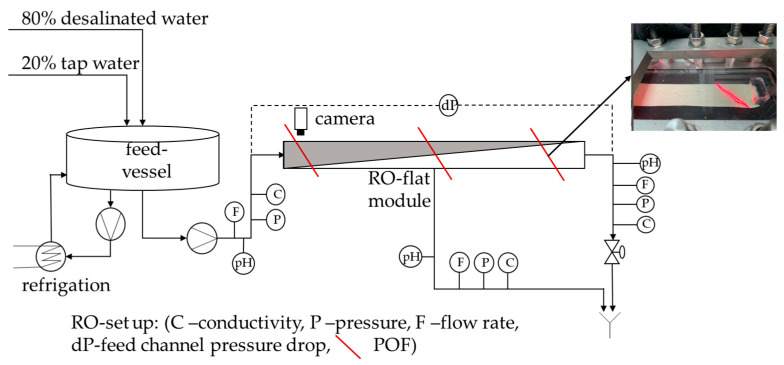
Schematic description of the reverse osmosis plant with integrated polymer optical fibers, sensors, and permeate and retentate rejection.

**Figure 2 membranes-13-00553-f002:**
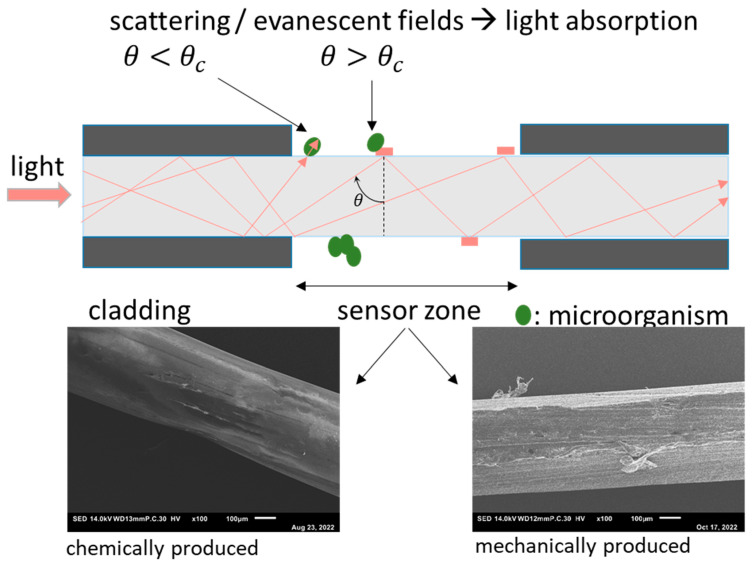
Mode of operation of the fiber-optical sensor. **Top**: schematic description of the optical fiber with removed fluoropolymer cladding and the two modes of interaction between light and deposits on the surface: scattering and evanescence-based attenuation. **Bottom**: scanning electron images (SEM) images of the sensor zone. In the left image, the cladding is removed using ethylaceteate, and in the right micrograph by mechanical abrasion.

**Figure 3 membranes-13-00553-f003:**
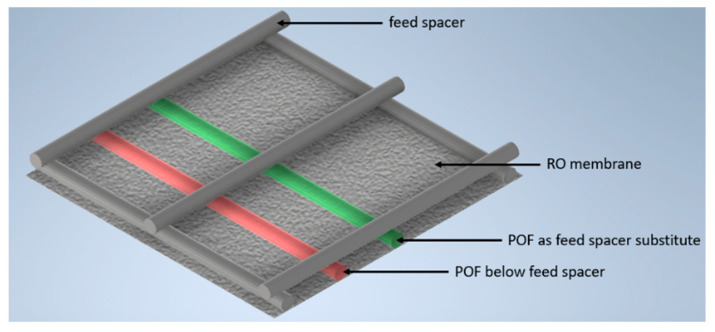
Depiction of the two polymer optical fibers that were included in a feed-spacer-square on the RO membrane surface to enable the detection of biofilm formation.

**Figure 4 membranes-13-00553-f004:**
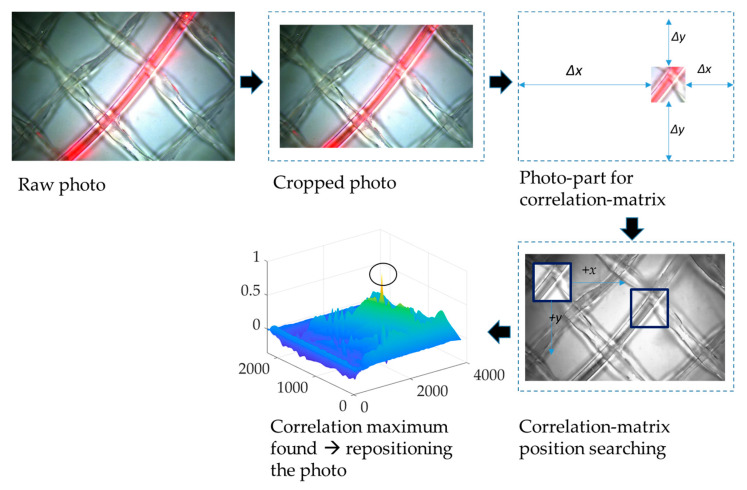
The process of image registration that was used for image analysis via the four steps of image registration described by Zitová and Flusser.

**Figure 5 membranes-13-00553-f005:**
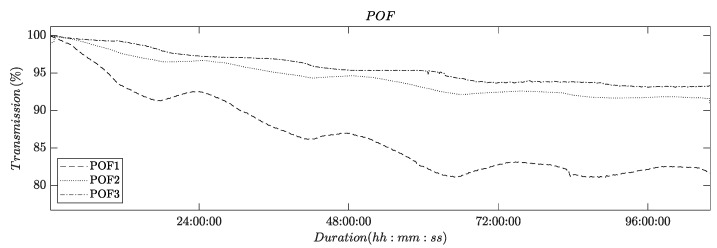
Plot showing the light transmission of POFs as a function of time. The feed in the experiment was an aqueous NaCl solution (2 g/L).

**Figure 6 membranes-13-00553-f006:**
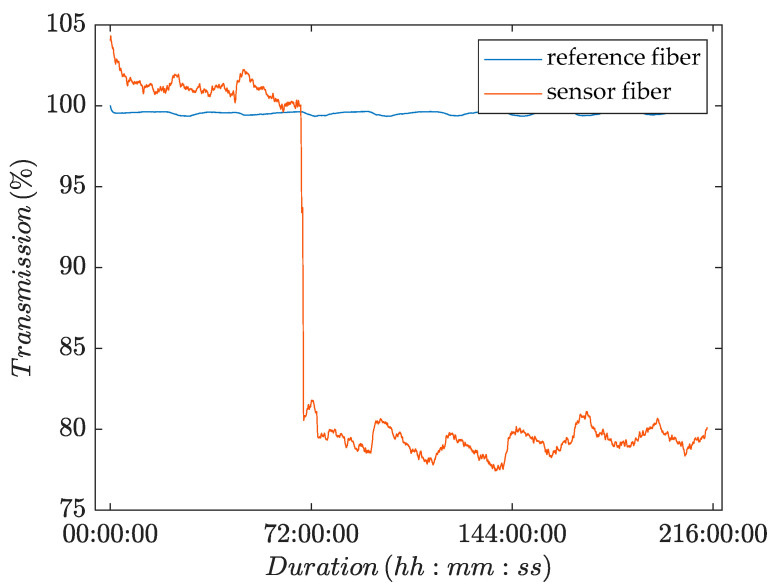
Change in transmitted light intensity as a function of time for POF sensors that were immersed in a water bath, to which yeast was added after a conditioning period of 71 h.

**Figure 7 membranes-13-00553-f007:**
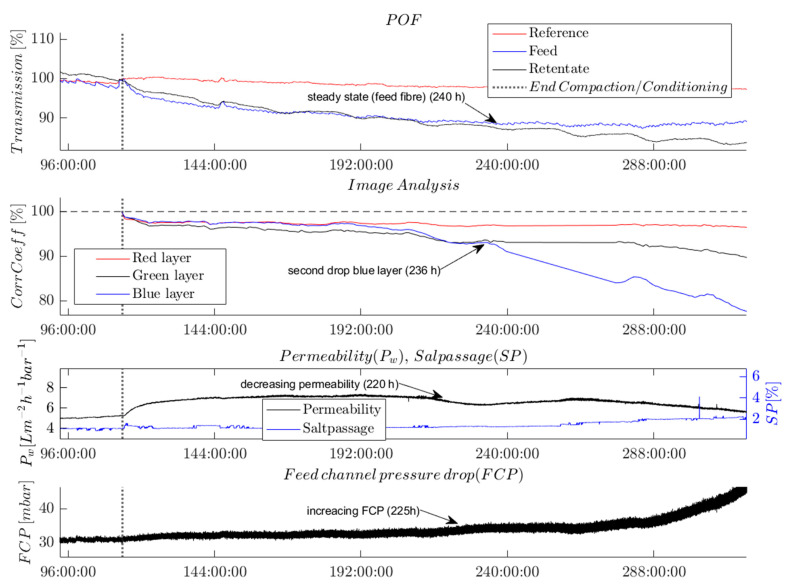
Time series showing the evolution of different selected online parameters. Experimental conditions: dosed nutrients (500, 100, 50) (C:N:P) in feed water (20% tap water, 80% desalinated water) from 114 h onwards.

**Figure 8 membranes-13-00553-f008:**
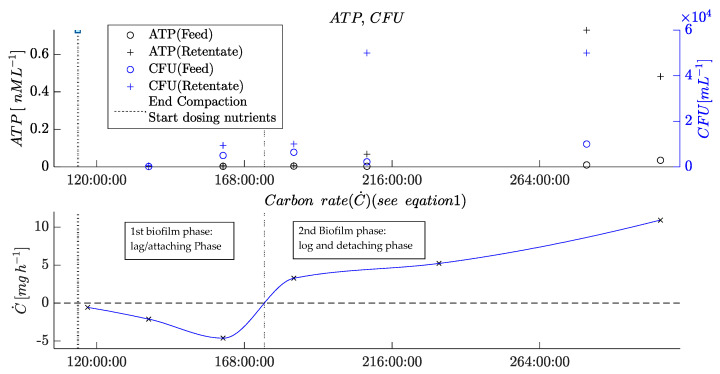
Plots of offline parameters of the biofouling experiment (see Figure 7).

**Figure 9 membranes-13-00553-f009:**
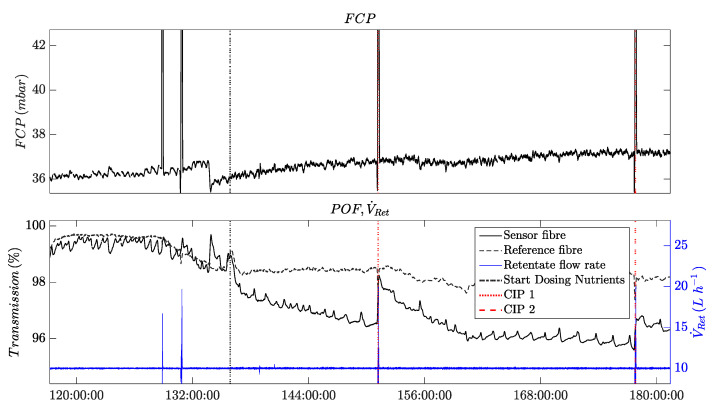
Plot of a time series from an experiment with cleaning-in-place (CIP) achieved by higher cross-flow velocities on an RO-Membrane with a POF sensor for monitoring and control.

**Figure 10 membranes-13-00553-f010:**
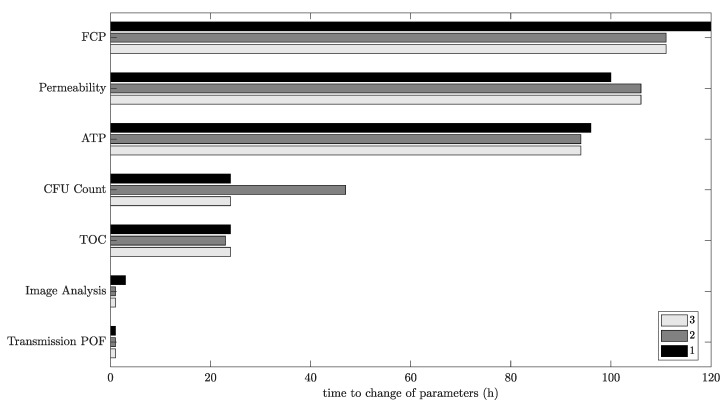
Comparison of parameter changes during the triplicate of biofouling experiments in the RO plant. Time starts when dosing nutrients to the tap water/desalinated water mixture.

**Table 1 membranes-13-00553-t001:** Relative abundances of MOs from a 16S rRNA analysis of a biofilm on the RO membrane after a 3-week experiment. Not italic: comments to the MO.

Phylum	Class	Order	Family	Genus
*Proteobacteria*	*Gammaproteobacteria* (typical biofilm formers)	*Burkholderiales*	*Commamonadaceae*	48% *Aquabacterium* 29% *Acidovorax* 3% *Delftla*
			*Rhodocyclaceae* *Burkholderiaceae*	2% *Ferribacterium* 2% *Cupriavidus*
		*Xanthomonadaceae*		3% *Pseudoxanthomonas* 0.8% *Stenotrophomonas*
		*Pseudomonales*		4% *Acinetobacter* 1% *Pseudomonas* (human pathogenic)
	*Alphaproteobacteria* (typical biofilm formers)	*Caulobacteraceae*		6% *Caulobacter* 0.7% *Phenylebacterium*
		*Sphingomonadaceae*		0.8% *Sphingopyxis*
				1% *Rhodobacter*
	*Bacteroidia*			0.8% *Cytophaga* (Coexisting within potable water biofilms [31])

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
