# Peer review of "Incipient Biofouling Detection via Fiber Optical Sensing and Image Analysis in Reverse Osmosis Processes"

_membranes, 2023, doi:10.3390/membranes13060553_

Round 1

Reviewer 1 Report

This research article entitled Incipient biofouling detection via fibre optical sensing and image analysis in reverse osmosis processes proposed a method to detect the membrane biofouling in membrane system. This method is innovative that can detect the sediment on the membrane effectively and prevent the influence of the sediment on the membrane in time. I recommend the paper to be published in the “Membranes” journal after solving the following questions.

1.        The manuscript should be polished by an editor whose native language is English to avoid grammar and spelling mistakes. For example, in line 14, it should be “is a major concern” instead of “are a major concern”. Similar flaws should be corrected and the manuscript should be revised carefully.

2.        The abstract should be re-written to make it more attractive. The present version is not well structured and is not interesting.

3.        Some absolute statement should be avoided, for example, the authors stated that “Early detection and removal is the only option for effective sanitation and prevention of biological growth in RO-spiral wound modules” in line 14-16. I think it is not suitable to use the word “only” since there are many other methods reported in literatures.

4.        In the introduction, please highlight the innovation of this paper and the advantages of the methods used. Some recent papers should be considered: Rational electron tunning of magnetic biochar via N, S co-doping for intense tetracycline degradation: Effciency improvement and toxicity alleviation. Enhanced energy recovery from landfill leachate by linking light and dark bio-reactions: Underlying synergistic effects of dual microalgal interaction.

5.        There are no error bars for all figures. I suggest the authors to add the error bar to increase the reliability of the manuscript.

6.        In the lines 89-90, please indicate the citric acid concentration to be used.

7.        In the lines 97-100, please state the concentrations of sodium acetate, sodium nitrate and sodium phosphate used as nutrients respectively.

8.        Figure 4 is too fuzzy. Please adjust the resolution appropriately.

9.        In the lines 253-257. Why does the screw pressure loss in the membrane plane module affect the transmittance of POF? Please analyze and explain with specific evidence.

English writting should be improved. 

Author Response

Answers to reviewer A

for the article entitled “Incipient biofouling detection via fibre optical sensing and image analysis in reverse osmosis processes

We thank the reviewer for reading the article and giving comments to our research.

  1. The manuscript should be polished by an editor whose native language is English to avoid grammar and spelling mistakes. For example, in line 14, it should be “is a major concern” instead of “are a major concern”. Similar flaws should be corrected and the manuscript should be revised carefully.

Answer We thank the reviewer for his/her comment and positive assessment of our original submission. We have carefully edited the manuscript for grammar and spelling and hope that the revised version meets the expectations.

  1. The abstract should be re-written to make it more attractive. The present version is not well structured and is not interesting.

Answer: We have carefully edited the abstract to make the impact of our study clear and hope that the reviewer feels that the revised abstract places our results more clearly in light of the current state-of-the-art.

  1. Some absolute statement should be avoided, for example, the authors stated that “Early detection and removal is the only option for effective sanitation and prevention of biological growth in RO-spiral wound modules” in line 14-16. I think it is unsuitable to use the word “only” since many other methods are reported in literature.

Answer: To address the issue raised by the reviewer, we have carefully revised the manuscript to avoid making any critical statements.

  1. In the introduction, please highlight the innovation of this paper and the advantages of the methods used. Some recent papers should be considered: Rational electron tunning of magnetic biochar via N, S co-doping for intense tetracycline degradation: Effciency improvement and toxicity alleviation. Enhanced energy recovery from landfill leachate by linking light and dark bio-reactions: Underlying synergistic effects of dual microalgal interaction.

Answer: We thank the author for these suggestions. We have edited the manuscript to highlight the advantages of the developed methods. The revised manuscript now also cites the publication “Rational electron tuning of magnetic biochar via N, S co-doping for intense tetracycline degradation: Efficiency improvement and toxicity alleviation”, which indeed provides a helpful reference that highlights the potential of catalytic oxidation processes in water treatment. We, however, do not see a connection between our study and the results reported in “Rational electron tunning of magnetic biochar via N, S co-doping for intense tetracycline degradation: Effciency improvement and toxicity alleviation”. Therefore, we decided to refrain from referencing this unrelated work.

  1. There are no error bars for all figures. I suggest the authors to add the error bar to increase the reliability of the manuscript.

Answer: The nature of the measurements would in this case render a presentation through error bars confusing. Key parameters are not comparable across independent experiments, but the trends of the observed development of the parameters over time provide insights into the formation of biofilms on the investigated membranes. We therefore chose to show triplicate experiments in full rather than condensing the data into averages with standard deviation. The qualitative results and the comparison of the triple determination are shown in the revised manuscript in Figure 9. We hope that this explanation addresses the concern of reviewer B.

  1. In the lines 89-90, please indicate the citric acid concentration to be used.

Answer: The concentration of all used chemicals for the CIP experiments has been added in the revised manuscript. The concentration of citric acid during the initial cleaning was 4 wt%.

  1. In the lines 97-100, please state the concentrations of sodium acetate, sodium nitrate and sodium phosphate used as nutrients respectively

Answer: As stated above, the concentrations of all chemicals have now been added to the revised manuscript. The used concentrations of sodium acetate, sodium nitrate, and sodium phosphate were included in the paragraph of the manuscript.

  1. Figure 4 is too fuzzy. Please adjust the resolution appropriately.

Answer: We thank the reviewer for this suggestion. Accordingly, the resolution of the figure has been adjusted in the revised manuscript.

  1. In the lines 253-257. Why does the screw pressure loss in the membrane plane module affect the transmittance of POF? Please analyze and explain with specific evidence.

Answer: We thank the reviewer for pointing this out. To explain this effect, we have carefully revised the corresponding paragraphs in the manuscript (see page 8). In contrast to the experiments carried out in the RO module, the yeast experiment described in lines 253-257 of the original manuscript was performed without a membrane flat module. In this case, no screws are needed, but the POFs were installed in a water bath into which a yeast suspension was added. The only affect on light transmission arises due to minor changes in swelling behaviour and temperature that are accounted for by maintaining the fibres in the water bath throughout the initial conditioning phase.

Author Response

Answers to reviewer B

for the article entitled “Incipient biofouling detection via fibre optical sensing and image analysis in reverse osmosis processes

We thank the reviewer for reading the article and giving comments to our research.

 The manuscript reports an interesting approach to detect and remove the foulants from RO membranes. The conclusions were sufficiently backed by experimental data. In my view, the work can be published after extensive language improvements. The current language has extensive feelings of non-academic presence.

Answer: We thank the reviewer for his/her comment and positive assessment of our original submission. We have carefully edited the manuscript for grammar and spelling and hope that the revised version meets the expectations.

Some observations are:

  • Some statements are very strong such as P1, line 14-15 (Early detection and removal is the only option for effective sanitation and prevention of biological growth in RO-spiral). It is not the only option. There are several other options.

Answer: We thank the reviewer for this suggestion. Accordingly, we have revised the manuscript to avoid such absolute statements.

  • Example of a non-academic terms such as P1, line 45 “Already in 1997 Flemming et al.”

Answer: Accordingly, we removed such non-academic terms.

Language improvement of overall manuscript to remove the confusing statements such as P2, line 49-50 “Hence, the crucial requirement in any anti-fouling strategy must is an early in-situ detection of microbial attachment”

Answer: We checked the manuscript, to avoid confusing statements and to improve the English.

Round 2

Reviewer 1 Report

Since the authors have revised the comments point by point. I recommend to be published in the present version.